# Molecular and Physiological Mechanisms to Mitigate Abiotic Stress Conditions in Plants

**DOI:** 10.3390/life12101634

**Published:** 2022-10-19

**Authors:** Baljeet Singh Saharan, Basanti Brar, Joginder Singh Duhan, Ravinder Kumar, Sumnil Marwaha, Vishnu D. Rajput, Tatiana Minkina

**Affiliations:** 1Department of Microbiology, CCS Haryana Agricultural University, Hisar 125004, India; 2Department of Biotechnology, Ch. Devi Lal University, Sirsa 125055, India; 3ICAR-National Research Centre on Camel, Bikaner 334001, India; 4Academy of Biology and Biotechnology, Southern Federal University, 344090 Rostov-on-Don, Russia

**Keywords:** microbiota, natural farming, physical factors, physiological changes, signal transduction and stressed conditions

## Abstract

Agriculture production faces many abiotic stresses, mainly drought, salinity, low and high temperature. These abiotic stresses inhibit plants’ genetic potential, which is the cause of huge reduction in crop productivity, decrease potent yields for important crop plants by more than 50% and imbalance agriculture’s sustainability. They lead to changes in the physio-morphological, molecular, and biochemical nature of the plants and change plants’ regular metabolism, which makes them a leading cause of losses in crop productivity. These changes in plant systems also help to mitigate abiotic stress conditions. To initiate the signal during stress conditions, sensor molecules of the plant perceive the stress signal from the outside and commence a signaling cascade to send a message and stimulate nuclear transcription factors to provoke specific gene expression. To mitigate the abiotic stress, plants contain several methods of avoidance, adaption, and acclimation. In addition to these, to manage stress conditions, plants possess several tolerance mechanisms which involve ion transporters, osmoprotectants, proteins, and other factors associated with transcriptional control, and signaling cascades are stimulated to offset abiotic stress-associated biochemical and molecular changes. Plant growth and survival depends on the ability to respond to the stress stimulus, produce the signal, and start suitable biochemical and physiological changes. Various important factors, such as the biochemical, physiological, and molecular mechanisms of plants, including the use of microbiomes and nanotechnology to combat abiotic stresses, are highlighted in this article.

## 1. Introduction

Stress is defined as a stimulus that inhibits the growth of plants and their metabolism and development at the time of both abiotic and biotic stress [1]. Abiotic stresses, such as higher or insufficient water supply, low and high temperature, heavy metals, ultraviolet radiation and salinity, are damaging to plant development and growth, and cause considerable losses in agricultural productivity worldwide [2,3]. When the stress threshold is exceeded, the plant is stressed, followed by activation of physiological, biochemical, morphological, and molecular-level mechanisms. The activation of these mechanisms can show the development of a fresh physiological state and the restoration of homeostasis in plants [4]. Due to abiotic stress, it has been estimated that crop production yield decreases by up to 70% in several commercially significant crops and they execute at just 30% of their genetic makeup in terms of yield [5].

Plants survive in environments that are repeatedly changing and frequently not suitable for plant growth as well as development. These harsh situations for the growth and development of plants arise mainly due to abiotic stress [6]. Moreover, abiotic stresses are anticipated to intensify and occur more frequently in the near future due to climate change, which may cause severe salinization of more than 50% of soil of the arable domain by 2050 [7,8]. Therefore, productive agricultural land and crop yields may gradually decline due to increasing temperature and recurrent flooding over several decades, particularly in the mid-latitudes [7,8].

Along with these factors, anthropogenic activities may increase the amounts of contaminants in the water, soil and air, with which plants must contend. It has been estimated that abiotic stress factors have a greater than 90% impact on plants and crop growth during their growing season in rural areas [9]. On the other hand, the rapid growth of the population has increased the demand for food and other necessary resources. Therefore, to develop stress-resistant cultivars that can endure abiotic stress and feed the expanding population, knowing plants’ stress responses is crucial. When plants faces variety of stresses, activates the stress signal and respond accordingly [9]. Plants with abiotic stress have primary signals for ion toxicity detection, low proline and chlorophyll content, low CO_2_ assimilation, and osmotic effects, etc., in the cells (Figure 1). Secondary consequences of these abiotic stresses are complicated and comprise oxidative stresses that harm various cellular components such as nucleic acids, proteins present in membranes and lipids, and metabolite malfunction. Thus, different abiotic stresses have distinctive and overlapping signals [10]. 

Drought and salt stress affect water potential homeostasis and distribution of ions at cellular as well as molecular levels. Alterations in water and ion homeostasis can cause growth inhibition, molecular damage, and even death [10]. Primary stress signals trigger some cellular responses; however, the rest are triggered by secondary stress signals. One of the vital features of these signals is the hyperosmotic signal, which increases phytochrome and abscisic acid in plants and provides a protective role during different abiotic stresses such as drought and salt stress [10]. Plants facing cold or chilling stress first indicate a change in cell membrane structure that affects the plant development, then disrupts protein or protein complex stability and lowers the ROS scavenging enzyme activity. These mechanisms cause photo-inhibition, decreased photosynthesis, and considerable membrane damage [10,11,12]. In addition, stress triggers protein synthesis and gene expression, as it triggers RNA secondary structure formations [13]. All these components of plant activity are critical for stress tolerance for minimizing the internal damage in the new stress environment so that homeostatic conditions must be reestablished and growth must be restored, though at a slower rate [5].

Considerable achievements have been made towards the knowledge of plant cellular and molecular mechanisms under different stress conditions [14]. Recognition of a stress environment causes changes in the expression of genes, resulting in changes in the composition of the plant proteomes, transcriptomes, and metabolomes. Plants’ response to different abiotic stress is not a simple pathway; however, it is an intricate integrated circuit comprising several pathways and precise tissue and cellular compartments and their interactions with additional cofactors, as well as signaling molecules for management of a particular response to a current stimulus. Thiourea (TU), a synthetic plant growth regulator with a composition of 36% nitrogen and 42% sulfur, has received much interest for its participation in plant stress tolerance. Thiourea helps to modulate some of the pathways associated with plants’ resistance to abiotic stress. Understanding the processes during TU-induced tolerance may help improve crop production under stress situations [15].

E3-ubiquitin ligases control abiotic stress responses either positively or negatively. Additionally, the target protein and the outcome of the changes in UPS-mediated breakdown, activity regulation, or translocation depend on the involvement of ubiquitin ligases-E3 enzymes in plants’ abiotic stress behavior. Consequently, recognizing and depicting ubiquitin ligases aims is vital during any stress response investigation [16]. The latest developments in understanding the molecular mechanisms underlying plant behaviors to abiotic stress highlight the complexity of these mechanisms, which involve a variety of processes including sensing, signal activation, transcription with transcript processing, and translation and posttranslational changes for combating the abiotic stress situations in plants (Table 1). The improved knowledge and use of various approaches, including genetic, chemical, and microbial techniques, enhance crop production and agricultural sustainability [17].

## 2. Abiotic Stresses and Crop Plants

Generally, various stresses act simultaneously, such as combined water, heat, salt, heavy metals, and other light stresses. As a result, these changes interfere with the regular plant metabolism function and the source–sink interaction, which lowers plant growth, metabolism, and production [55]. Moreover, these stresses alter the expression sequence of several plant genes and have huge effects on crop production worldwide, reducing the usual yields of important crops such as wheat and rice [16,56].

### 2.1. Salt Stress

Soil salinity is a huge risk for agriculture in areas where water shortage and poor drainage systems of irrigated farms lower the productivity of crops significantly [57]. According to the Food and Agricultural Organization (FAO 2016) [58], more than 6% of total global land and 19.5% of total irrigated land is already affected by salt conditions. Both human and natural factors can cause soil salinity. Of a total 932.2 Mha salt-affected soils globally, 76.6 Mha soil salinization has been caused by human beings [59]. The salt-affected lands having higher amounts of either exchangeable sodium or soluble salts, both perhaps due to insufficient leaching of cations that forms the base. The chief soluble salts that act as anions are sulfate (SO_2_^−4^SO_4_^−^), carbonate (CO_2_^−3^CO_3_^−^), chloride (Cl^−^) and nitrate (NO^−3^NO_3_^−^) salts and the cations are potassium, calcium, magnesium, and sodium. Many metals, such as selenium, lithium, boron, strontium, silica, fluorine, rubidium, manganese, aluminum, and molybdenum are present in hyper-saline soil water, some of which can be harmful to human, animal, and plant health [6,57].

Salt accumulation slows down the growth of plants and lowers plants’ absorption capacity for nutrients and water, a consequence of osmotic stress. The level of resistance to salt stress changes from species to species. Cereal crops such as barley can resist up to 250 mM NaCl; moderately salt-resistant crops are bread wheat, maize, sorghum, while wheat is less resistant to salinity [60]. Plant development is reduced by subsequent salt exposure during two phases: ionic toxicity and osmotic stress [61].

### 2.2. Drought Stress

Rainfall distribution is unequal due to climate change, which is responsible for the major stress reported: drought, which is the most prevalent abiotic stress globally, reducing grain output drastically. It has a devastating impact on the ability to fulfill the food requirements of the increasing worldwide population. Drought stress is linked to a lack of water and cellular dehydration. A decrease in water potential, stomatal closure, and turgor pressure results in poor plant growth and development [62]. Low-water stress affects biochemical and physiological functions, including ion acquisition and chlorophyll synthesis, photosynthesis, respiration, and carbohydrate and nutrient metabolism, resulting in reduced plant growth [63]. Plant adaptation to low-water stress is a process that involves physiological and biochemical alterations in the plant system. Under drought conditions, plants limit their shoot development and metabolic demands. In maize, yield reduction is observed up to 40%, and in wheat, 21%, with around 40% water shortage or reduction [64]. A yield decline ranging from 34 to 68% was reported in cowpea during drought stress [65].

### 2.3. Cold Stress

This stress has been identified as the main abiotic stress, which reduces agricultural crop productivity by decreasing crop quality and postharvest life. Cold stress consists of chilling from 0–15 °C and freezing at 0 °C, negatively affecting plant growth and agriculture production [66,67]. During the comparison of both freezing and chilling stress, it has been found that freezing stress is far more harmful to plants. Typically, freezing’s harmful effects start with the formation of a nucleation of ice within the cells, then progressively grows and forms ice crystals, causing water leakage and cell dehydration [68,69]. Several major crops are still unable to cope with cold acclimation. For example, tomato (*Solanum lycopersicum*), maize (*Zea mays*), rice (*Oryza sativa*), cotton (*Gossypium hirusutum*), and soybean (*Glycine max*) cannot withstand lower temperatures and have the capacity to grow and survive only in tropical and subtropical areas [70]. Hence, cold stress has a negative impact on plant growth along with plant metabolism and development, resulting in the reduction of crop yields globally [69,71].

### 2.4. Heat Stress

During heat stress, plants are highly sensitive; at extremely high temperatures, plants die. Generally, plants perform better at their optimum temperature; below and above the optimum temperature plant growth and development are severely affected [72]. Most biochemical and enzymatic reactions double at temperatures from 20 °C to 30 °C and change with every 10 °C. Temperatures above and below the optimum range reduce the reaction rate because enzymes are denatured and inactivated progressively. One or two degrees of temperature change have a huge impact on plant growth and development, particularly reproduction, causing a negative impact on early stages of male gametophyte formation in various crops, including wheat, rice, sorghum, barley, maize, and chickpea [5,73]. Heat stress causes male sterility and spikelet production abnormalities in rice and wheat [5]. In wheat and rice, cold and heat stress results in tapetum breakdown, and changes in the callose walls of microspores, exine formation and metabolism of carbohydrate, ultimately ensuing in male sterility [5,74]. However, temperature stress shows no adverse effects on female gametophyte development [75].

### 2.5. Heavy Metals Stress

Metal poisoning is one of the main environmental risks that impairs plants’ ability to operate and engage in normal metabolic activities. Heavy metals (HMs) are a group of non-biodegradable, persistent inorganic substances having an atomic weight of more than 20 and a density of more than 5 g cm^−3^, which affect and pollute the food chains, irrigation, soils, aquifers, and nearby atmosphere before having mutagenic, cytotoxic, and genotoxic effects on human, plant, and animal health [76,77].

Toxic metals are accumulated in the agricultural soils due to excessive use of chemical fertilizers along with increasing industrialization, showing harmful consequences to the soil–plant interaction system [78]. These metals concentrate and enter the plant system at a slow rate via water and air and enter the food chains over a certain time [79]. This poses a considerable threat to the natural food web and biogeochemical cycle [80]. The unprecedented in vivo heavy metals accumulation and bioaccumulation in the environment presents a dilemma for all plants and organisms. Toxic concentrations of HM can interact with various important cellular molecules, including nucleoproteins and DNA, causing excessive production of ROS [6,81]. This will result in serious plant changes, e.g., proteolysis, shoot chlorosis, and lipid peroxidation [80,82]. Under abiotic stress, it was thought that using osmolytes, nanoparticles, mineral nutrients, hydrogels, antioxidants, protectants, potassium, and plant growth hormones such as uniconazole and salicylic acid would boost plant production [83,84]. Additionally, plants may adapt to the negative impacts of droughts by applying plant hormones such as brassinolide (BR), gibberellic acid (GA), auxins, ABA, cytokinins, JA and ethylene, which govern several beneficial reactions in plants [83,84].

## 3. Sensing and Responding Mechanisms of Plants during Abiotic Stress Conditions

Biological molecules that act as sensors detect undesirable environmental changes and elicit quick stimuli to abiotic stress by signal molecules that activate the system. Abiotic stress causes additional Ca^2+^ to enter the cytoplasm from apoplastic sources. Ca^2+^ entrance passages are one kind of sensor for detecting stress signals [85,86,87,88]. The Ca^+^, nitric oxide, and reactive oxygen species (ROS) also work as messenger substances that activate plant responses during cold stress. Reactive oxygen species such as hydrogen peroxide, hydroxyl radicals, and superoxides are formed in response to various kinds of stress [89]. Receptor-like kinases have an intracellular and extracellular domain at which ligand binding occurs and protein-with-protein binding will occur. When a sensor protein attaches to the extracellular domain, the histidine residues found in the intracellular domain self-phosphorylate and are activated. The activated ligand or sensor proteins may then induce signal-specific cellular response via a MAPKs cascade. Intracellular signaling, i.e., protein dephosphorylation and phosphorylation, regulate a broad range of cellular functions: enzyme activation, macromolecule assemble, protein localization, and degradation [90,91]. When plants detect abiotic stress, signaling cascades are activated, followed by activation of kinase cascades, the assembly of ROS and plant hormones accumulation, leading to the induction of precise set of genes responsible for combating the plant abiotic stress, as shown in Figure 1.

The metabolism of cytokinins, ABA, and ethylene are all impacted by stress, and these molecules then interact with certain kinases to control various biological functions, from controlling shoot development under stress to stomata movement [92]. Abiotic stress in plants, e.g., low water conditions (drought stress), low temperature, and excessive salinity stimulate the expression of the huge range of genes present in plants. With gene expression, different proteins are formed in different plant parts that prevent damage to the cell and activate the large number of genes essential for several abiotic resistance processes in plants. Various kinds of proteins are produced, such as chaperones and late embryogenesis abundant proteins (LEA proteins), which are primarily involved in the development of tolerance. At the same time, stress-associated genes are all concerned with generating the stress response [93]. Plant genes responsible for stress are regulated at three stages, i.e., transcriptional, posttranscriptional, and posttranslational.

### 3.1. Gene Regulation at the Transcriptional Level 

Transcriptional regulation, consisting of chromatin modification and remodeling enhancers and promoters, has regular binding sites positioned downstream and upstream of the coding area known as cis-regulatory and trans-regulatory elements, typically transcription factors. Various abiotic stresses cause alterations in the methylation pattern of histone proteins and DNA, which repress or promote gene transcription. Promoters are unique sequences that have a regulatory role, binding RNA polymerase and other transcription factors to initiate transcription [94]. C-repeat binding factors (CBF), dehydration responsive element binding (DREB), MYB, zinc finger families, and leucine zipper (bZIP) are examples of the regulatory elements concerned in plant defense systems along with genes responsible for stress during binding of the responsive gene promoter’s cis-element [95]. Drought tolerance was improved by *Oryza sativa* WRKY 11 (trans-regulatory element) overexpression under heat shock protein 101 (HSP 101) promoter control [96,97]. A significant discovery is new cis-acting elements, C-repeat and DRE, which respond to low temperature, drought (low water stress), and excess salt stress [98]. C-repeat binding factor proteins have been isolated progressively since their discovery by identification of DNA-associated proteins which attach the DRE and CRT motifs [99]. C-repeat binding factors 1–3 are cold-induced CBF genes found in *Arabidopsis* and are located on chromosome IV in tandem; CBF1-3 include Apetala2 or ethylene responsive type transcription factors, which directly bind with the CRT/DRE-conserved motifs present in the promoters of CBF regulons, also known as COR genes, and trigger gene expression during a low-temperature environment [99,100]. Transgenic *Arabidopsis* with CBF1 overexpression has more COR expression and is more resistant to freezing [101]. Orthologous expression of CBFs has been reported in various plants types such as tomato, wheat, rice, maize, and barley, along with heterologous expression of CBFs in *Arabidopsis*, which also improves their freezing tolerance mechanism [102], and suggests that it shows tolerance to cold only in those plants with CBF genes from tomato only. Hence, it is reported that CBF1-3 play an essential role in modulating cold tolerance and it is not more conserved in every species, but also it is species-specific [102].

### 3.2. Gene Regulation at Posttranscriptional Level

Posttranscriptional gene regulation refers to the regulation that happens between the stages of pre-mRNA and mRNA translation. It involves four levels: (A) pre-messenger RNA processing via capping as well as splicing along with polyadenylation; (B) nucleoplasmic trafficking of mRNA; (C) mRNA turnover and stability; (D) translocation of mRNA [103]. Alternative splicing is another strategy that plays an essential role in gene regulation during heat and cold stress, e.g., a gene stabilized 1 (STAT1) encoding a nuclear pre-mRNA provides cold resistance in *A. thaliana* [18,104]. Cold responsive (COR) gene regulation is essential for posttranscriptional regulation in plants. A dead box gene expression regulator (RCF1) plays an important role in the correct pre-mRNA splicing of various COR genes during low-temperature stress [105]. It has been investigated that alternate splicing pathways alter gene expression to cope with temperature stress [106]. Small RNA segments consisting of 20 to 25 nucleotides are formed from non-coding dsRNA precursors via dicer-like (DCL) RNases, generating several posttranscriptional gene silencing processes. One of these processes, mediated by 21 nucleotide microRNAs, cleaves mRNAs or blocks their translation [107].

### 3.3. Gene Regulation at the Posttranslational Level

Protein phosphorylation, ubiquitination, and sumoylation are the posttranscriptional activities that play an essential role in modifying plant behavior to many abiotic stresses. SNF1-related protein kinase (SnRKs) and MAPKs known as mitogen-activated protein kinases are key players in various signal transduction cascades initiated by osmotic stress and dehydration via phosphorylation of particular residues [108]. They include XERICO and SnRK2 gene codes for an H2-type zinc finger, and E3-ubiquitin ligases, which are associated in ABA-dependent response during water stress, such as stomata closure [109].

Posttranslational histone modifications and DNA methylation are linked to gene expression changes in response to chilling or cold stress. Histone protein acetylation and deacetylation are due to histone acetyl transferase (HAT) and histone deacetylases (HDAs) being involved in the plant during cold stress [110]. HAD 6 of *Arabidopsis* is overexpressed due to low temperature and positively enhances the freezing resistance [111]. During low-temperature stress, HDAs come into sight directly with maize DREB1 gene activation and histone hyper-acetylation. It has been reported that histone acetylation of ZmDREB1Aas well as ZmCOR413 in maize and OsDREB1 gene in rice histone is activated by lower temperature [112,113]. In the case of *Arabidopsis*, the RNA-mediated methylation 4 (RMP4) protein was found to play an important role in RNA-directed DNA methylation by combining with RNA polymerase Pol II and Pol V [114]. During cold stress, gene RDM4 is essential for Pol II possession at CBF2 and CBF3 gene promoters to fight abiotic stress in plants [115].

## 4. Crucial Signal Transduction Mechanism for Abiotic Stress

Abiotic stresses, which induce signal transduction pathways, such as drought, cold, light, heat and salt, are classified into three types. The first one is MAPK modules, which are used in osmotic/oxidative stress signaling to create antioxidant substances, ROS scavenging enzymes, and osmolytes; the second is calcium-dependent signaling, which promotes the triggers of LEA-type genes; followed by calcium-dependent SOS (salt overlay sensitive), which maintains ion homeostasis, as shown in Figure 2.

### 4.1. Oxidative or Osmotic Stress Signaling in Plants

All stresses involving salt, heat, drought, oxidative, and cold stress cause the formation of ROS species and cause significant damage to plants [116]. A greater level of ROS functions as a signal and one of the preventive plant responses is the production of ROS scavengers. Osmotic stress triggers various protein kinases, such as MAPKs, in restoring osmotic homeostasis. As a result, osmotic stress activates sensor or receptor proteins such as G protein-coupled receptor proteins, tyrosine and histidine kinases, which activate the MAPK network and signaling cascade, and are associated with the production of more osmolytes, required during osmotic stress. The primary function of osmolytes in cell turgor pressure maintenance is to act as a driving force for water uptake. Well-suited solutes such as proline, glycine betaine, mannitol, and trehalose will serve as ROS and chemical chaperones by stabilizing membrane proteins [117].

The MAP kinase network modulates and transmits the signals from the cell surface to the nucleus. Three kinases are triggered consecutively by the upstream kinase in the core MAPK cascades. The MAP kinase phosphorylates on threonine and serine residues after the activation of MAPKKK, and once activated the MAPK either transfers to the nucleus directly and triggers the transcription factor, stimulating other signal substances to regulate gene expression to cope with stress [118].

### 4.2. Ca^2+^-Dependent Activation of LEA Genes

Abiotic stress increases with calcium ions’ entry into the cytoplasm. Calcium ion entry channels serve as sensors for abiotic stress detection. Calcium ions trigger CDPKs (calcium-dependent protein kinases) and are represented by multigene families; their expression stages are modulated both geographically and temporarily during complete development. The CDPK pathway plays an important role in the production of large numbers of anti-desiccation proteins via the activation of LEA-type genes, indicating the damage and repair processes, which are distinct from the pathways that regulate osmolyte synthesis [119,120]. Seeds are naturally desiccated during maturation to minimize desiccation shock at the time of germination by accumulating high levels of LEA proteins [121]. Water deficiency, low temperature, excessive osmolarity, and low-temperature stress cause crop plants to accumulate LEA protein. These proteins are utilized to protect proteins from denaturation or renaturation, maintain membrane integrity and protein structure, and sequester ions in affected tissues. Many scientific publications state that chaperones and LEA proteins protect macromolecules against dehydration, such as lipids, enzymes, and mRNA [122,123]. LEA proteins specialize in membrane desiccation defense, while antioxidant enzymes and compounds have a role in desiccation tolerance. Both LEA proteins and osmolytes work in association with membrane structure and protein stabilization by conferring favored hydration during moderate desiccation conditions and changing water levels to protect against abiotic stress in plants [124].

### 4.3. Calcium Ion-Dependent SOS Signaling

Plants response to high salt concentration, change the ion transporter channel, helps in the regulation of ion homeostasis during salinity. Higher intracellular or extracellular sodium ions function during the SOS pathway, primarily increasing a cytoplasmic calcium ions signal, which affects the expression and activity of different ion transporters, including K^+^, sodium ions, and H^+^. A shift in turgor is used as the input for osmotic stress signaling. Salt stress signaling pathways including osmotic and ionic homeostasis signaling routes, detoxification pathways, and growth maintenance pathways are reported in *Arabidopsis* [125,126]. During abiotic stress signaling, evidence shows that CDPKs and SOS3 of calcium ion sensors have a main role in coupling an inorganic signal with a specified protein phosphorylation pathway and seem important for plant salinity resistance [127].

## 5. Functions of the Microbiome in Abiotic Stress Management

Plants are not isolated entities and are not able to live alone. Instead, they coexist with various microbes, including bacteria, fungi, protists, viruses, and other microorganisms [128]. These microorganisms coexist in various plant tissues, forming the plant’s microbiome, which lives in three different places: the phyllosphere, endosphere, and rhizosphere [127]. It is becoming increasingly proven that mycorrhizal fungi and useful soil bacteria, including PGPB and PGPR, have a significant role in sustainable farming and agriculture by stimulating plant growth and increasing plant tolerance to abiotic stresses [127]. Many microbes are essential in plant development, metabolism, and growth under abiotic stress situations (Table 2; Figure 3).

### 5.1. Arbuscular Mycorrhizal Fungi (AMF)

The development and health of plants may be impacted by AMF, which are symbiotic soil-borne fungi [128]. Agroecosystem services related to AMF still have significant knowledge gaps that need to be filled to be optimized, even though they are currently thought to have significant potential stability and sustainability in the agriculture system, and huge progress has been reported in the understanding of the arbuscular mycorrhizal symbiosis. 

It has been found that AMF plays an essential role during abiotic stress, when various stresses are combined [129]. Recently, the role of AMF for protection aligned with abiotic stress in tomatoes has been reported [130]. In conjunction with halophytes and glycophytes, AMF can protect against salt stress [131]. A researcher [132] demonstrated a contrary higher dependence of glycophytes versus AMF compared to halophytes during salt stress. The advantages of AMF under salt stress were the subject of meta-analyses of various genes required for stress tolerance [130,133] (Auge et al. 2014, Chandrasekaran et al. 2021). The osmolyte, carbohydrate, and antioxidant systems can all be improved by AMF [134]. A high K^+^/Na^+^ level is maintained when they are in symbiosis with plants, preventing the absorption or transfer of harmful Na^+^ [135]. 

AMF can also affect how efficiently carbon is used by maintaining larger grain yields, faster rates of stomatal conductance, net photosynthesis, and poorer internal water use efficiency during salinity stress [136]. AMF-associated mechanisms have recently been discussed in [137], ranging from the uptake of nutrients to increased water use efficiency, from osmoprotectant and ionic homeostasis to enhanced photosynthetic efficiency, from cell structure protection to strengthening functions along with triggering antioxidant metabolism, till phytochrome profile modulation. AM-colonized plants show a modification in plant metabolism, specifically, an enhancement in proline amount with greater H_2_O_2_ and isoprene emission in contrast to not-inoculated plants [135]. In the context of water stress conditions, AMF also has shown a beneficial outcome in tomato plants; however, the effects varied depending on the features and fungi species taken into account [138,139]. When three AMFs from various genera were examined to see how they affected tomato resistance to salt or drought stress, it became clear that all studied AMFs shared certain responses; however, others were unique to individual isolates [140].

**Table 2 life-12-01634-t002:** Microbiome-mediated abiotic stress resistance and mechanism in plants.

Type of Microbes	Abiotic Stress Type	Plant and Tolerance Mechanism	References
*Pseudomonas putida* P45	Drought	Sunflower (*Helianthus annuus*) showed EPS production and enhanced soil aggregation	[141]
*Pseudomonas*	Drought	Pea (*Pisum sativum*) plants reduced the production of ethylene	[142]
*Azospirillium sp.*	Drought	Wheat (*Triticum aestivum*) has better water relations	[143]
AM fungi	Drought and salinity	Sorghum (*Sorghum bicolor*) showed better water relations	[144]
*Scytonema*	Coastal salinity	Rice (*Oryza sativa*) with extracellular products and gibberellic acid	[145]
*Burkholderia phytofirmans*	Cold	Grapevine (*Vitis vinifera*) with ACC-deaminase synthesis	[146]
*Burkholderia* sp. and *Methylobacterium oryzae*	Cd and Ni toxicity	Tomato (*Solanum lycopersicum*) with lower uptake along with translocation of heavy metals	[147]
*Pseudomonas fluorescences*	Salinity	Groundnut (*Arachis hypogaea*) with ACC-deaminase synthesis	[148]
*Rhizobium tropici*; *P. polymyxa*	Drought	Common bean (*Phaseolus vulgaris*) with change in hormonal composition and stomatal conductance	[149]
*Glomus intraradices* and *Pseudomonas mendocina*	Drought	Lettuce (*Lactuca sativa*) has antioxidant status improved	[150]
*Pseudomonas* strain AMK-P6	Heat	Sorghum (*Sorghum bicolor*) with better biochemical status due to activation of heat shock proteins	[151]
*Glomus* sp. and *Bacillus megaterium*	Drought	*Trifolium* (*Trifolium repens*) with proline and IAA production	[152]
*Paraphaeosphaeria quadriseptata*	Drought	*Arabidopsis* (*Arabidopsis thaliana*) has HSP-heat shock protein induction	[153]
*Bacillus subtilis*	Salinity	*Arabidopsis (Arabidopsis thaliana)* has reduced root Na^+^ import by reduced transcriptional expression of AtHTK1 (a high-affinity KC transporter) genes	[153]
*Pseudomonas putida*	Salinity	Cotton (*Gossypium hirusutum)* stopped the salinity-associated accumulation of ABA in seedlings	[154]
*Glomus etunicatum* and *Glomus clarum*	Salinity	Wheat (*Triticum aestivum),* Chilli (*Capsicum annum*) and mung bean (*Vigna radiata)* have increased KC concentration in root and reduced NaC in shoots and root	[155]
PGPRs	Heat	Clover (*Trifolium repens*) plants with greater nitrogen fixation	[156]
*Bacillus licheformis*	Drought	*Capsicum annum* with expression and activation of stress-related proteins and genes	[157]
*Bacillus thuringiensis*	Drought	Wheat (*Triticum aestivum)* showed the organic compound production	[158]
*Pantoea dispersa* and *Azospirillium brailense*	Salinity	*Capsicum annuum* has an increase in photosynthesis rates well as stomatal conductance	[159]
*Burkholderia phytofirmans* and *Enterobacter* sp.	Drought	Maize (*Zea mays*) showed an increased rate of shoot and root biomass	[160]
*Pseudomonas koreensis* strain	Salinity	Soyabean (*Glycine max*) has increase KC level and decreased Na^+^ level	[161]
*Enterobacter intermedius*	Zn toxicity	White mustard (*Sinapis alba*) with ACC deaminase, IAA IAA, hydrocyanic acid, and solubilization of phosphate	[162]
*Serratia* sp. and *Bacillus cereus*	Drought	Cucumber *(**Cucumis sativa)* showed the production of genes responsible for the synthesis of proline, an antioxidant enzyme, and monodehydroascorbate	[163] Wang et al.; 2012
*Photobacterium* spp.	Mercury toxicity	Common reed (*Phragmites australis*) showed activity of IAA and mercury reductase	[164]
*Rhizobium leguminosarum* and *Pseudominas brassicacerum*	Zinc toxicity	Mustard (*Brassica juncea*) with metal chelating molecules	[165]
*Rhizobium*	Salinity	Asian rice (*Oryza sativa*) with RAB 18 salt stress-associated gene expression	[166]
PB 50 strain of *B. megaterium*	Drought	Rice (*Oryza sativa*) showed better plant growth under osmotic stress, plants protected via stomatal closure with enhanced soluble sugar, carotenoid content and protein content	[167]
*Bacillus albus* and *Bacillus cereus*	Drought	Maize (*Zea mays*) seeds have a higher germination rate and increased seedling length with reduced toxic effects	[168]
*Gluconacetobacter diazotrophics* (Pal5)	Drought	Rice (*Oryza sativa* L.) shows gor, P5CR, BADH and cat genes expression with increase glycine betaine and proline content	[169]
*Penicillium* sp. and *Calcoaceticus*	Drought	Foxtail millet (*Setaria italica*) with increased glycine betaine and proline content, sugars and chlorophyll a and b with the decrease in lipid oxidation	[170]
*Streptomyces pactum* and *actinomyces*	Drought	Wheat (*T. aestivum*) reduces stress via an enhancement in sugar levels and antioxidant enzymes	[171]
*B. amyloliquefaciens; Pseudomonas putida*	Drought	Chick pea (*Cicer arietinum*) with better photosynthesis, chlorophyll content, biomass and osmolyte content	[172]
*PGPR consortium*	Salinit	Common bean (*Phaseolus vulgaris*) with available iron content in the soil increased	[173,174]
*B. gladioli*; *P. aeruginosa*	Cd toxicity	Tomato (*Solanum lycopersicum*) with higher expression of metal transporter genes	[175]
*Mesorhizobium; Rhizobium*	Salinity	Chickpea (*Cicer arietinum*) with enhanced nitrogen fixation	[176]
*Bacillus megaterium*	Osmotic	Maize (*Zea mays*) with higher expression of 2 ZmPIP isoforms in roots	[177]

Additionally, in drought circumstances, AMF has valuable effects on maize, promoting plant development and photosynthesis by considerably increasing mineral absorption, assimilation and chlorophyll content, compatible solute concentration, and triggering the antioxidant defensive system [129]. The structures and functions of the photosystem PSII and PSI are less likely to be harmed by water stress due to AMF, as has also been shown [178,179,180]. The latest research discovered that the stimulation of fungal genes’ encoding for SOD scavenging enzymes and non-enzymatic defenses such as glutaredoxin, metallothioein, etc., had a protective function against ROS burst [181]. The contrary impacts of AMF on aquaporin expression and plant hormone levels rely on the type of fungus, aquaporin, and applied stress [182]. Since dryness and salt cause generalized stressors, it stands to reason that AMF, which enables plants to survive under high-salinity environments, makes plants resistant to drought [182]. Transcriptomics has helped understand how several crop species, including rice [182,183], tomatoes [184]), grapevines [185] (Balestrini et al. 2017), as well as wheat [186], regulate expression of fungal as well as plant genes in AM interactions. 

Transcriptomic analysis has recently been used in tomato roots colonized by AM to confirm the contribution of the AM symbiosis to tomato plant response, nematode function, and water stress, and revealed unique information regarding the response to AM symbiosis. A function for arbuscular mycorrhizal colonization in initiating defensive behavior against root-knot nematodes was also suggested by alterations in the tomato gene expression associated with nematode attack during AM symbiosis [187]. 

### 5.2. Actinomycetes

Several studies have shown *Actinomycetes* to thrive in various stressful environments, including drought, high temperatures, and high salt. They significantly reduce the harmful effects of abiotic stresses while encouraging plant development [188]. Streptomyces play a synergistic role in wheat crops under severe abiotic conditions and promote plant development and their capacity to withstand such conditions [189]. Furthermore, because of their unusual shape, they mix with the soil particles in the rhizosphere and create a solid link with the plants. This approach assists the plant in strained soil and enables more effective utilization of water and nutrients in the rhizosphere. These bacteria increase their capacity to withstand abiotic stresses by an array of mechanisms, comprising changes in root and cell wall morphology, 1-ACC deaminase activity, and the ability to protect from oxidative damage with the production of proline and glycine betaine, which aid in osmoregulation processes [190]. It has been found that wheat seeds soaked with streptomyces inoculum showed greatly enhanced shoot length, root depth, and high root and shoot biomass, and appreciably boosted the germination of roots under high-saline conditions [191]. Tomato plants inoculated with *Streptomyces* sp. PGPA39, during salt stress, showed a substantial reduction in leaf proline concentration and enhancement in plant biomass compared to the non-treated plants [192].

Treating maize plants with Actinomycetes during low-water conditions produced plants with higher growth, survival rate, shoot and root dry weight, and chlorophyll content than untreated plants [193]. Furthermore, Hasegawa and colleagues [194] demonstrated that endophytic *Actinomycetes* II improves drought resistance in *Kalmia latifolia* L. (mountain laurel) by causing callose buildup and lignification in the cell wall. According to next-generation sequencing methods used for microbial communities’ identification, it has been investigated that Actinobacteria have regularly been discovered as one of the bacteria most found in soils [195]. However, although actinomycetes have played an essential role in reducing abiotic plant stress, slight knowledge regarding the dynamics of their interactions with plants limits the potential for using these microorganisms in agriculture.

### 5.3. PGPR/Plant Growth-Promoting Bacteria

The plant rhizosphere is a highly delicate and dynamic ecosystem, with different housing types of microorganisms that play various roles in plant development and survival [196]. Numerous distinctive and illustrative PGPR bacteria, comprising the *Bacillus*, *Pseudomonas*, *Enterobacter*, *Agrobacterium*, *Klebsiella*, *Azotobacter*, *Ervinia*, *Bradyrhizobium*, *Burkholderia*, and *Serratia*, have been isolated and described [197,198]. Conversely, several variables, including plant growth stage, species, cultivation method, soil ecology, and testing settings may affect how effective PGPR treatment is at enhancing plant growth. Rhizobacteria that promote plant development can lessen the drought stress effects and increase yield in treated plants [199,199]. It has been reported that *Rhizobacterium* induces drought endurance and resilience, known as RIDER, which results in biochemical alterations, and allows PGPR to decrease the drought stress effects [200,201]. Phytohormones development, bacterial exopolysaccharides formation, cyclic metabolic pathway conventions, and optimization of the antioxidant defense system are associated with the deposition of various carbon-based substances, including amino acids; polyamines, sugars, and heat shock protein production are examples of RIDER mechanisms [202,203].

Recently, it has been investigated that PGPB, along with other plant microbiomes, acts as a biological technique for decreasing plant salt stress [173]. *Rhizobacteria* that encourage plant development can successfully lessen the effects of drought on *Zea mays* and wheat grass. Two bacterial strains, namely *Enterobacter* sp. 16i and *Bacillus* sp. *12D6*, were employed to counteract drought stress in plants and it was reported that *Bacillus* sp. *12D6* performed better under drought stress [204]. *Azospirillum* (GQ255950)-treated maize plants perform better and have more root and shoot fresh weight biomass along with proline, soluble sugars, amino acids, and osmotic levels compared to non-treated maize plants [205]. A significant biotic stress element that reduces agricultural output is drought. In prior research, *Pseudomonas libanensis* EU-LWNA-33 was used to treat wheat plants, and measurements of growth parameters showed that the root length and biomass were enhanced after treatment. Additionally, biochemical analysis revealed that at 75% stress, levels of proline increased up to two times while levels of glycine betaine were enhanced 1.2-fold [206].

Plant survival under various types of stress is challenging and depends specifically on the plant’s root microbiome. It has been demonstrated that bacteria isolated from harsh environmental circumstances contain traits that make them resistant to salt stress. *P. fluorescens* was isolated from Saharan rhizosphere soil and showed a PGPB property in maize under salt stress. Numerous microorganisms living on plants with high salt content exhibit adaptations to salinity stress and do well in these environments [207]. One study was conducted in a greenhouse where the soil was treated with three isolated bacteria from rice fields. The soil showed how drought stress might change microbial interactions, and soil microbial interactions with plants were changed with water limitation, e.g., *Actinobacteria* and *Chloroflexi* were abundant in the changing patterns, while *Acidobacteria* and *Deltaproteobacteria* were lost. Plant survival under drought circumstances results from compartment-specific reorganization [208].

Several methods of PGPB might respond to drought stress effects and promote phytohormones as well as solute production, chlorophyll synthesis, and activation of 1-aminocyclopropane-1-carboxylate deaminase, exo-polysaccharides, and mineral solubilization [209]. *Bacillus amyloliquefaciens* GB03 and *Pseudomonas flurescencs* WCS417R- treated plants showed that activating the antioxidant defense system considerably decreased the effects of drought stress [210].

*Rhizobacteria* that encourage plant development, including *Enterobacter*, *Bacillus*, *Pseudomonas* and *Moraxella*, have been identified. Under drought stress, wheat plants treated with *Bacillus* species showed enhanced auxin production up to 25.9 g mL^−1^ along with an increase in field capability of up to 10% as well as increased crop output of 34% [211]. In Uttar Pradesh, India, PGPR strains including *Pseudomonas putida* and *B. amyloliquefaciens* have been identified in alkaline soils, and the combined impact was studied in chickpea plants, which produced plants with boosted biomass, photosynthesis, osmolyte, and chlorophyll content, and reduced abiotic stress, as well as being eco-friendly to the environment [172]. Wheat seedlings treated with *Azospirillum brasilense* Sp245 had enhanced water control, greater mineral content, higher amounts of K, Mg, and Ca, and a 12.4% overall improvement in grain yield when compared to untreated plants [143].

Foxtail millet crops were used in an experiment, treated with *Penicillium* and *Acinetobacter*, which revealed increased physiological growth and a reduction in the harmful effects of drought stress [170]. During this study, buildup of proline, osmolytes, and glycine betaine with elevated levels of chlorophyll content was reported that helped reduce drought stress [170].

## 6. Nanoparticles’ Application in Combating Abiotic Stress in Plants

Numerous plant responses are triggered by abiotic stress, ranging from growth and morphological changes to crop output and yield [212]. Nanotechnology is one of the newest and most promising methods for treating abiotic stress conditions in plants, such as HMs stress, heat stress, drought stress, and salinity stress. It is also an environmentally friendly technique. With the increasing cellular antioxidants, nutrient uptake, photosynthetic efficiency, and molecular as well as biochemical pathways, NPs dramatically increase the ability of plants to withstand abiotic stress [213]. Modernizing the agricultural sector with prospective applications for improving plant growth as well as development under stress conditions, nanotechnology has recently made strides [213].

Several studies have been reported on the possible use of NPs in the remediation of HM-contaminated soil [214,215,216]. A study of FeO NPs found they can relieve drought stress [217,218] and reduce Cd toxicity in wheat plants by enhancing biomass, chlorophyll levels, and antioxidant enzymes [216]. Si NPs have been reported to reduce HM-induced phytotoxicity in wheat as well as rice and pea [219,220]. To lessen the detrimental effects of HMs on plant growth and development, new nanoremediation techniques must be developed. 

It has been found that Si NP treatment enhances plants’ ability to withstand drought stress [221]. By modifying the morpho-physiological characteristics of barley plants, Si NPs showed promising drought stress recovery [222]. It has been revealed that in saline- and water-deficient circumstances, Si NPs improved cucumber growth and yield [223]. Wheat plants under drought stress exhibited higher relative water content, CAT, SOD activity, yield, and biomass after application of chitosan [224]. Silver nanoparticles were used to lessen the harmful effects of drought in lentil (*Lens culinaris* Medic.) plants [225]. Abscisic acid administration aided by Si NPs was described as an efficient management technique to increase drought resistance in *Arabidopsis thaliana* [226].

Wheat plants’ growth, chlorophyll concentrations, and antioxidant enzymes were improved by treatment with FeO NPs, which reduced the effects of salt stress [216]. It has been found that manganese NP seed-priming controls salt stress by altering the molecular reactions in pepper (*C. annuum* L.) plants [227]. To better understand how NPs work to increase plants’ resistance to salinity and other abiotic stress in plants, more molecular and physiological research is required. Numerous studies have demonstrated the possible use of NPs to increase the ability of plants to withstand heat stress [228,229]. Recently it has been reported that the use of organically produced Se NPs (100 g/mL) enhanced wheat development by enhancing plant tolerance to heat stress [230]. Researchers have discovered that silver nanoparticles considerably improved the morphological characteristics of wheat plants under heat stress conditions [231]. Overall, using metallic NPs as nanofertilizers can help plants be more resilient to heat stress, which is important for sustainable agriculture. 

## 7. Conclusions

Evidence from various studies regarding molecular, biochemical, and physio-morphological characteristics, plant responses to diverse types of abiotic stresses, and the microbe interaction mitigation methods in plants have been examined. These studies have improved our knowledge of the processes behind gene cascades, microbial interactions, and metabolic pathways, along with augmentation of different proteins, enzymes, metabolites, and the down- and upregulation of numerous genes. This paper provides dynamic information about plants’ response to different types of abiotic stress. This paper gives innovative ideas for improving the current methods to mitigate abiotic stress by employing various gene regulation-based and microbial-mediated plant interactions, facilitating improved germination, increased ability to withstand and mitigate adverse environment conditions, and better yield in plants. By supplying vital nutrients, nanotechnology can be a cost effective and promising way to increase plants’ ability to withstand abiotic stress. However, because of their widespread use, there are some possible worries regarding their adverse impacts on the environment.

Plants have evolved built-in adaptation mechanisms to deal with various complex abiotic challenges. With the aid of science and technological advancements, it is now feasible to comprehend gene function, establish gene-manipulation strategies, and generate plant characteristics to combat abiotic stress. Signaling pathways must be seen as intricate networks. Hence, abiotic stress signal-transduction cascades are better understood by molecular investigations of the signaling molecules. Abiotic stresses have a detrimental effect on plants that is exacerbated by ongoing climate change, reducing agricultural production. Abiotic stress tolerance is a multigenic response that involves stress-responsive gene expression, signal transduction, and sensing. As a result of these coordinated efforts, agricultural production, productivity, and food security will improve.

## 8. Future Perspectives

Genetic modification and recombinant techniques should be integrated with traditional and marker-assisted breeding practices in the future to produce plants with modified characteristics that cope with adverse conditions. In addition to this, nanotechnology will be the most potent tool for combating the abiotic stresses in plants. Moreover, finding new adaptable germplasm is crucial for directing breeding programs to identify plants for a changing climate and future research should focus on developing NPs that are inexpensive, nontoxic, self-degradable, and eco-friendly using green methods. By addressing the effects of climate change, particularly stress caused by drought, heat and cold, these combined initiatives will significantly advance the cause of food security through increased agricultural output and productivity.

## Figures and Tables

**Figure 1 life-12-01634-f001:**
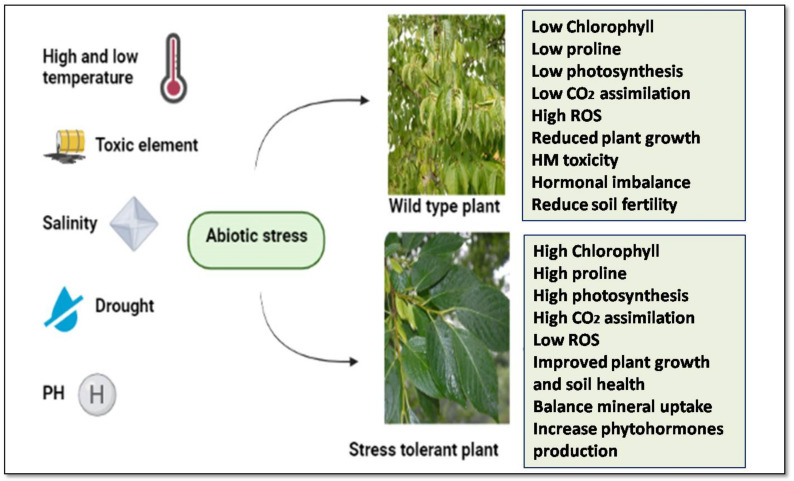
Plant behaviors during abiotic stress conditions.

**Figure 2 life-12-01634-f002:**
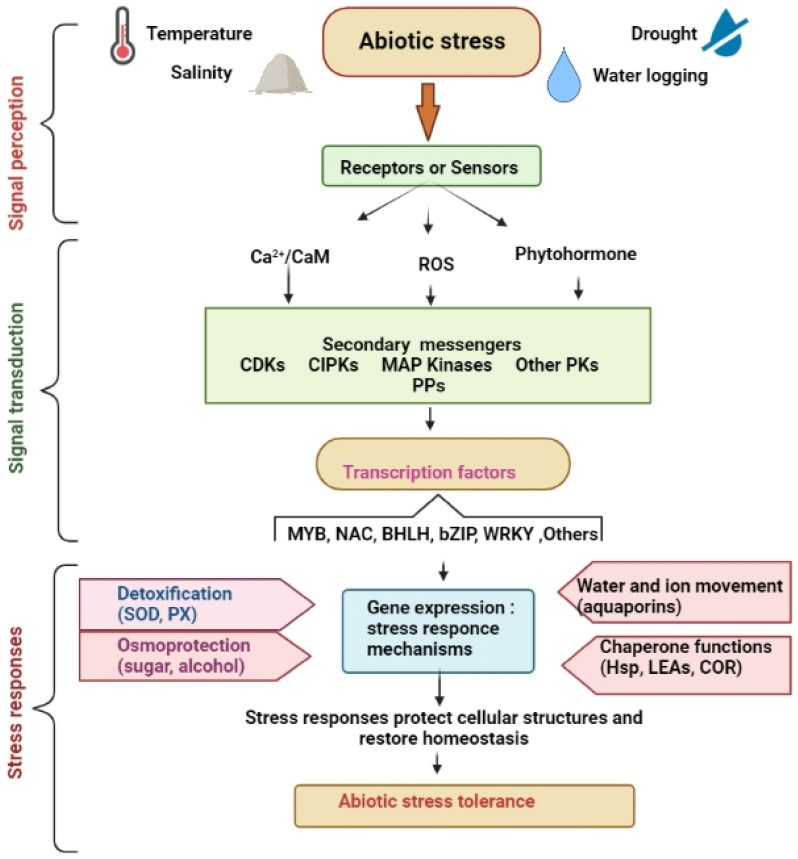
Signaling mechanisms for abiotic stress tolerance in plants.

**Figure 3 life-12-01634-f003:**
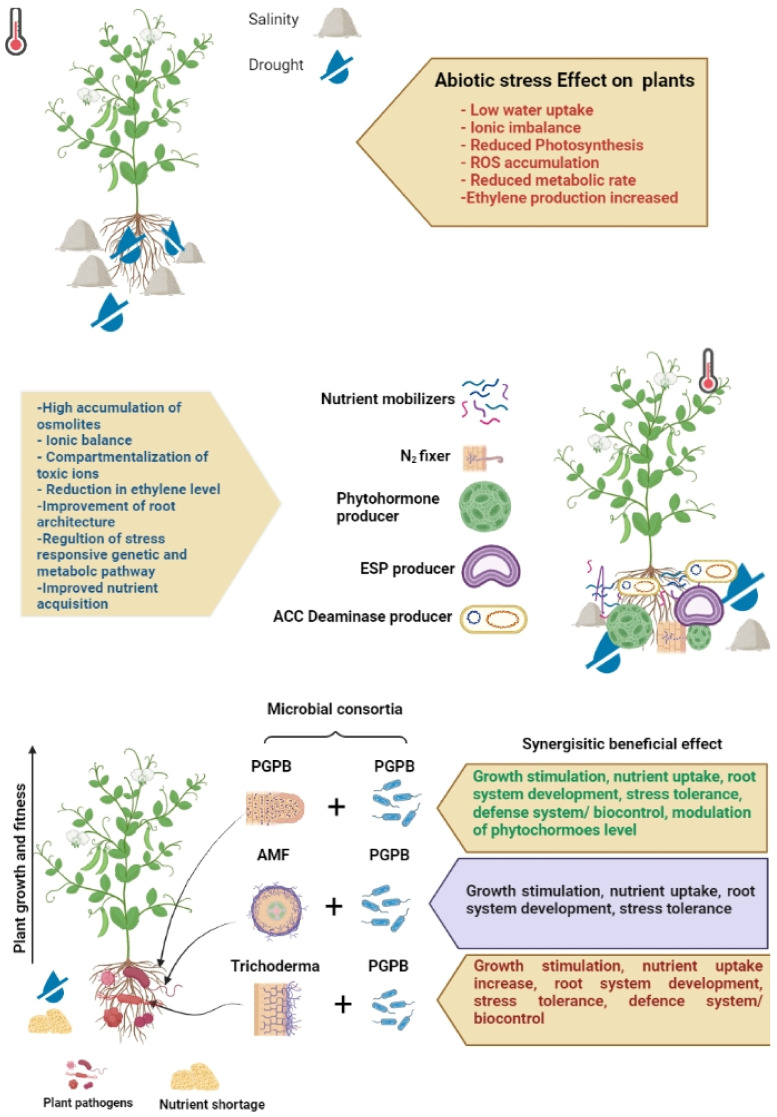
Functions of different microbes in combating abiotic stress conditions.

**Table 1 life-12-01634-t001:** Different studies concerning abiotic stress tolerance in plants.

Abiotic Stress Type	Mechanism and Key Parameters Studied	Plants/Crops	References
Cold stress	The CBF (C repeat binding factor) transcriptional cascade, CBF expression and CBF-independent regulons mediate the transcriptional regulation and pre-mRNA processing, export, and degradation involved in post-regulatory mechanisms	*Arabidopsis*	[18]
Low-temperature stress	Alter hormonal expression		[19]
Heat and drought stress	Enhancing the accumulation of carbohydrates		[20]
Heat stress	Autophagy plays a vital role in cellular homeostasis, metabolism, and other processes		[21]
Heat stress	ceRNA networks are mediated by the differentially expressed circRNA, by the influence of various important genes, and participate in response to hydrogen peroxide, heat stress, and phytochrome signaling pathway		[22]
Water stress	Lipid peroxidation decreases with scavenging reactive oxygen species and higher excitation energy dissolution due to photochemical quenching with reduced excitation pressure		[23]
Drought stress	The physiological activities and antioxidant protective systems modulate CarMT gene overexpression		[24]
Drought stress	H_2_S endogenous production rate increases and a noteworthy transcriptional reorganization of pertinent miRNAs		[25]
Drought stress	A transmembrane potassium ions efflux as well as calcium and chloride ions influxes are induced due to endogenous hydrogen sulfides		[26]
Cold and drought stress	Dehydrins concentrated in roots and stems	Blueberry	[27]
Heat stress	Lower accumulation of H_2_O_2_ and damage to cells	Strawberry	[28,29]
Salinity stress	Plant response is positively regulated due to OsH1RP1-ring finger protein 1	Maize	[30]
Cold stress	Changes in DNA methylation	Rice	[31]
Heat stress	Lipid peroxidation as well as antioxidant enzymes in roots and leaves		[32]
Drought stress	E3-ubiquitin breakdown		[16,33]
Heat stress	Candidate genes as well as quantitative trait loci		[34]
Cold stress	Linear electron transport chain is downregulated and PSII is repressed, as represented by the lowering in the PSII photochemistry efficiency along with electron transport efficiency	Hibiscus	[35]
Water deficit and heat stress	Contribution of ferredoxin-mediated cyclic pathway and chlororespiration		[36]
Salinity stress	Simultaneous expression of variable expressed genes		[37]
Cold stress	Enhances epidermal cell density, stomatal density and index, width of xylem vessel and phloem tissue and sclerenchyma	Candyleaf	[38]
Salt stress	Accumulation of biomass, ions concentration in tissue and steviolglycosides		[39]
Drought stress	The use of steviol glycosides enhances the harvest index		[40]
Salinity and drought stress	Sodium chloride serves as an activator, and mannitol works for the downregulation of genes involved in the steviol glycosides synthesis pathways that alter the steviol glycosides production		[41]
Cold stress	Photosynthetic electron transport chain protection by the sub-cellular antioxidant system	Wheat	[42]
Heat and drought stress	Signaling of phytohormone and epigenetic control		[43,44]
Salinity stress	Maintenance of osmoprotectants, photosynthetic activity and sodium/potassium ions ratio		[45,46]
Cold stress	WRKY gene expression	Grapevine	[47]
Heat stress	HSPs genes, along with antioxidant enzyme expression	Tomato	[48]
Salinity stress	Reduced the accumulation of different ions such as sodium, magnesium, and zinc in leaves and roots	Commonbean	[49]
Drought stress	Enhances plant metabolism with water relation parameters, antioxidant enzyme water relation parameters, activities of antioxidant enzymes and yield per plant increases	Lemon grass	[50,51]
Salinity stress	Antioxidants in leaves and lipid peroxidation	Tomato	[52]
Salinity stress	Biomass production as well as stomatal conductance		[53]
Drought stress	Sugar and amino acid content accumulation	Alfalfa	[54]

## Data Availability

Not applicable.

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
