# Peer review of "Molecular and Physiological Mechanisms to Mitigate Abiotic Stress Conditions in Plants"

_life, 2022, doi:10.3390/life12101634_

Round 1

Reviewer 1 Report

This review article deals with the mechanisms to mitigate abiotic
stress in plants. The article covers vast information
on an interesting subject and will be useful to the readers. The review is well presented, however, improvement is required in the language. The article can be accepted for publication after addressing these minor corrections. Pointing out here some specific suggestions:

  1. Authors should add concise summary above abstract.

  2. Abstract: Rewrite, should be crisp

  3. Page 1, Line 25-29- Rewrite

  4. Line 34: Replace adaptive with adoption; Line 40: Biochemical change as biochemical.

  5. Line 28 to 31 should be merged as it is dealing same aspect, even same meaning.

  6. Line 48-49 and 60-61- Repititive; Line 55 and 62-Repititive ; Line 100- toward-towards, plants-plant,s

  7. Line 103: Responses to different stress is not a simple pathway: Correct the grammar. Many such mistakes are there in MS, authors are advised to correct them. Excessive use of comma everywhere, correct that.

  8. E3-ubiquitin ligases and thiurea details are misfit in Introduction

  9. Line 134: No full stop, Line 135: add Full form of FAO; Line 234-237- Simplify

  10. Table 1: At many places, plants/crops are not mentioned, please specify. Table 2- In 2nd column (abiotic stress type)- Writing stress can be avoided every time. It can be written as drought, salinity etc. All figures text should be corrected such fig 1 Chlorophyll is written as colorophyll

  11. Pls add reference to support line 64-66. This is a factual information, thus, must be supported by literature.

  12. In introduction section, authors have not supported several facts by literature such as line 72-81. Careful attention should be paid to cite work properly.

  13. Scientific names must be added. Pls correct thorough out manuscript and follow uniform pattern.

  14. Future perspective should be separately added considering recent technology's role to enhance these molecular and physiological mechanisms to overcome the biotic stress. The more emphasize can be drawn on nanotechnology.

  15. Line 521: “Wheat plants, when plants treated with Bacillus species” : Revise

Author Response

Reply to Reviewer 1

Dear anonymous reviewer, authors are thankful for your positive evaluation and critical remarks on manuscript. We have addressed all the comments and supplied track change file where changes are marked. 

Sr. No.

Comments

Reply

1

Authors should add concise summary above abstract.

Added as suggested

2

Abstract: Rewrite, should be crisp

Modified

3

Page 1, Line 25-29- Rewrite

Rewrite as suggested

4

Line 34: Replace adaptive with adoption; Line 40: Biochemical change as biochemical.

 Replaced as suggested

5

Line 28 to 31 should be merged as it is dealing same aspect, even same meaning.

Merged as suggested

6

Line 48-49 and 60-61- Repititive; Line 55 and 62-Repititive ; Line 100- toward-towards, plants-plant,s

Corrected as suggested

7

Line 103: Responses to different stress is not a simple pathway: Correct the grammar. Many such mistakes are there in MS, authors are advised to correct them. Excessive use of comma everywhere, correct that.

 Modified as suggested

8

E3-ubiquitin ligases and thiurea details are misfit in Introduction

Both have important role in the abiotic stress management in plants

9

Line 134: No full stop, Line 135: add Full form of FAO; Line 234-237- Simplify

In Line 134- full stop added

add Full form of FAO- added

Line 234-237- Simplify- simplified as suggested

10

Table 1: At many places, plants/crops are not mentioned, please specify. Table 2- In 2nd column (abiotic stress type)- Writing stress can be avoided every time. It can be written as drought, salinity etc. All figures text should be corrected such fig 1 Chlorophyll is written as colorophyll

In table 1 we have arranged the studies according to plants and use the crops name only as single for many references by merging the different studies column in case of plant name

In table 2 modified as suggested

In figure 1 is modified as suggested

11

Pls add reference to support line 64-66. This is a factual information, thus, must be supported by literature.

Supportive ref. added

12

In introduction section, authors have not supported several facts by literature such as line 72-81. Careful attention should be paid to cite work properly.

Supportive ref. added

13

Scientific names must be added. Pls correct thorough out manuscript and follow uniform pattern.

14

Future perspective should be separately added considering recent technology's role to enhance these molecular and physiological mechanisms to overcome the biotic stress. The more emphasize can be drawn on nanotechnology.

Modified as suggested

15

Line 521: “Wheat plants, when plants treated with Bacillus species” : Revise

Revised as suggested

Reviewer 2 Report

The reason for this decision is:

This manuscript does not fulfill the standards established for the journal to be considered for publication.

This paper describes the molecular biological mechanism for abiotic stress. However, although Table 1 describes various studies on the abiotic stress tolerance of plants, the types of plant names are not accurately described, and the range of abiotic stress of plants is too narrow. In Figures 1 and 2, the results and signaling content of the response are too abstract. In particular, it is questionable what is the difference between Figure 3 and Figure 2. In addition, Table 2 describes the resistance and mechanisms of microbial community-mediated abiotic stress in plants, but it is not clear whether it is the abiotic stress of crops or the results of the biotic stress study. As suggested in the title of this text, there is no presentation of molecular biological or physiological experimental results for relieving abiotic stress.

Author Response

Reply to Reviewer 2

Authors thank for your positive evaluation and critical remarks on manuscript. We have addressed all the comments and supplied track change file where changes are marked. 

Sr No.

Comments

Reply

1

Table 1 describes various studies on the abiotic stress tolerance of plants, the types of plant names are not accurately described, and the range of abiotic stress of plants is too narrow.

Table 1 describes various studies on the abiotic stress tolerance of plants, the types of plant names are not accurately described – It looks like because In table 1 we have arranged the studies according to plants and use the crops name only as single for many references by merging the different studies column in case of plant name

The range of abiotic stress of plants is too narrow - At least 40 references is included in table 1.

2

In particular, it is questionable what is the difference between Figure 3 and Figure 2.

Figure 2 is based on molecular basis of Signaling mechanisms for abiotic stress tolerance in plants

And

Figure 3 is based on role of different microbes in combating abiotic stress conditions

3

Table 2 describes the resistance and mechanisms of microbial community-mediated abiotic stress in plants, but it is not clear whether it is the abiotic stress of crops or the results of the biotic stress study.

Table 2 clearly describes the Microbiome -mediated abiotic stress resistance and mechanism in plants

Reviewer 3 Report

Detailed revision was shown as follows.

-- 1. Introduction. The section should be reorganized according to a logical structure (e.g., abiotic stress types, sensing methods, response characteristics of plants).

-- Lines 76-77. In my opinion, it is insufficient to describe the sentence “Plants with abiotic stress have primary signals for ion toxicity detection and osmotic effects in the cells” through the Figure 1, especially for the ion toxicity detection and osmotic effects.

-- Some minor errors should be corrected, e.g., Line 101. A full stop is missing after the Ref. [14]. Line 238. The first letter should be capitalized for figure 1. Line 250. The i.e; should be i.e.

-- Why is the 20 ℃ selected for the Figure 3?

-- More suggestions and innovative ideas for improving the current methods to mitigate

abiotic stresses should be discussed and proposed in Section 2.

-- Lines 533, 535. Two ways of reference labels ([170] (Kour et al; 2020a).) are simultaneously used.

Author Response

Reply to Reviewer 3

Authors appreciate your kind remarks and valuable suggestions. We’ve addressed and supplied track change file for your kind perusal.

Sr No.

Comments

Reply

1

Lines 76-77. In my opinion, it is insufficient to describe the sentence “Plants with abiotic stress have primary signals for ion toxicity detection and osmotic effects in the cells” through the Figure 1, especially for the ion toxicity detection and osmotic effects.

Modified and corrected

2

Some minor errors should be corrected, e.g., Line 101. A full stop is missing after the Ref. [14]. Line 238. The first letter should be capitalized for figure 1. Line 250. The i.e; should be i.e.

Corrected as suggested

3

Why is the 20 ℃ selected for the Figure 3?

Modified as requirement

4

More suggestions and innovative ideas for improving the current methods to mitigate abiotic stresses should be discussed and proposed in Section 2.

In section 2 only types of abiotic stress is discussed not techniques

5

Lines 533, 535. Two ways of reference labels ([170] (Kour et al; 2020a).) are simultaneously used.

Corrected and modified

Reviewer 4 Report

Life-1971137 

Saharan  et al., submitted a review manuscript entitled “Molecular and Physiological Mechanisms to Mitigate Abiotic  Stress Conditions in Plants“ for publication consideration in Life.

This review aimed to provide information about abiotic stress, mainly drought, salinity, low and high temperature in plant and how plant response to. 

The authors tried to summarize what plant reaction mechanisms associated biochemical and molecular changes and emphasised some relevant microbiomes. 

The manuscript is well-written and relevant. 

It would be welcomed if the authors could change the title to reflect the research and work of microbiomes. Another point is related to “Simple Summary:” part, it should be dropped and I believe Life is not journal to argue about this suggestion. 

Author Response

Reply to Reviewer 4

The authors thank reviewers for their recommendations/suggestions and positive evaluation of our manuscript. The itemized responses are provided to address all the comments. The revised manuscript showed the track changes.  

Sr No.

Comments

Reply

1

The authors tried to summarize what plant reaction mechanisms associated biochemical and molecular changes and emphasised some relevant microbiomes. 

The summary for microbes is discussed in section 5  first paragraph

2

It would be welcomed if the authors could change the title to reflect the research and work of microbiomes. Another point is related to “Simple Summary:” part, it should be dropped and I believe Life is not journal to argue about this suggestion. 

Modified as suggested and Summary part is added

Round 2

Reviewer 2 Report

Although it is called a review paper, it has not been significantly modified from the last one submitted. And I think I wrote a review with very limited knowledge. We strongly reject organized scientific knowledge as it can be misleading.

Author Response

Dear anonymous reviewer 

Thank you for remarks! We've thoroughly revised whole manuscript and changes are marked in R1 and R2 track changes files. Due to your valuable remaks current version reached up to the clear readability and quality of presentation.